# SARS-CoV-2 S Mutations: A Lesson from the Viral World to Understand How Human Furin Works

**DOI:** 10.3390/ijms24054791

**Published:** 2023-03-01

**Authors:** Leonardo Cassari, Angela Pavan, Giulia Zoia, Monica Chinellato, Elena Zeni, Alessandro Grinzato, Sylvia Rothenberger, Laura Cendron, Monica Dettin, Antonella Pasquato

**Affiliations:** 1Department of Industrial Engineering, University of Padova, Via Marzolo 9, 35131 Padova, Italy; 2Department of Biology, University of Padua, Viale G. Colombo 3, 35131 Padova, Italy; 3European Synchrotron Radiation Facility, 71, Avenue des Martyrs, 38000 Grenoble, France; 4Institute of Microbiology, University Hospital Center and University of Lausanne, Rue du Bugnon 48, 1011 Lausanne, Switzerland; 5Spiez Laboratory, Federal Office for Civil Protection, Austrasse, 3700 Spiez, Switzerland

**Keywords:** Furin, protease, SARS-CoV-2, cleavage, peptide, in vitro, glycoprotein S, envelope glycoprotein, virus

## Abstract

Severe acute respiratory syndrome coronavirus-2 (SARS-CoV-2) is the etiological agent responsible for the worldwide pandemic and has now claimed millions of lives. The virus combines several unusual characteristics and an extraordinary ability to spread among humans. In particular, the dependence of the maturation of the envelope glycoprotein S from Furin enables the invasion and replication of the virus virtually within the entire body, since this cellular protease is ubiquitously expressed. Here, we analyzed the naturally occurring variation of the amino acids sequence around the cleavage site of S. We found that the virus grossly mutates preferentially at P positions, resulting in single residue replacements that associate with gain-of-function phenotypes in specific conditions. Interestingly, some combinations of amino acids are absent, despite the evidence supporting some cleavability of the respective synthetic surrogates. In any case, the polybasic signature is maintained and, as a consequence, Furin dependence is preserved. Thus, no escape variants to Furin are observed in the population. Overall, the SARS-CoV-2 system per se represents an outstanding example of the evolution of substrate–enzyme interaction, demonstrating a fast-tracked optimization of a protein stretch towards the Furin catalytic pocket. Ultimately, these data disclose important information for the development of drugs targeting Furin and Furin-dependent pathogens.

## 1. Introduction

December 2019 went down in history as the month when a new deadly virus was first identified. The initial spread flew under the radar until a significant number of cases of atypical pneumonia suddenly caught people’s eye. At that point, the entire planet witnessed the astonishing measures taken to curb the contagion across the population within the city of Wuhan, Hubei Province, Central China, which was the first site of the viral outbreak. Unfortunately, the attempts failed to limit the infection that quickly spread worldwide. As of September 2022, 600 million confirmed cases and 6.5 million deaths had been reported by the World Health Organization (WHO) (https://covid19.who.int/). The virus responsible for the pandemic was identified as a novel member of the *Coronaviridae* family and named as severe acute respiratory syndrome coronavirus-2 (SARS-CoV-2). The WHO called the disease associated with SARS-CoV-2 coronavirus disease-19 (COVID-19), and it was the virus first reported in the year 2019 [1].

SARS-CoV-2 is thought to have emerged from bats, possibly via a secondary host [2]. Several factors have been crucial for the global human pandemic, including—but not limited to—the acquisition of specific mutations that allowed binding to the human receptor [3] and the insertion of key amino acids that made the virus maturation dependent on human ubiquitous proteases [4]. Multiple other aspects need to be taken into consideration and have contributed to the spread and the severity of the viral infection of SARS-CoV-2. For example, age of population, concomitant diseases [5], and variations in human lifestyle (such as diet), along with microbiota, which play a crucial role in health and immunity against viral infections [6,7]. Since the very first infection and human-to-human propagation, SARS-CoV-2 has rapidly co-evolved with its host. In the initial stages of the pandemic, viral evolution was shaped by selection imposed by a naïve host and the environment, resulting in new variants with adaptive advantages rapidly taking over previous strains. The scenario changed following the advent of vaccines, resulting in the increase in immune-evading variants [8,9,10,11].

SARS-CoV-2, as with all viruses, is an opportunistic agent, which relies on the infected host for its replication and dissemination. Since the very first contact with the target cell, and the delivery of its genetic material and thereafter full cell colonization, SARS-CoV-2 manipulates and re-programs the host to its advantage [12]. A key step of the virus life-cycle is the synthesis and maturation of the protein spike S, which sits on the surface of budding particles [13]. Only S-decorated virions are competent for cellular receptor(s) engagement and membrane fusion [14]. The glycoprotein S is made of 1273 amino acids, is membrane anchored, and heavily undergoes glycosylation and further cleavage, leading to the mature heterodimer S1/S2. The entire synthesis and post-translation modifications occur at the expense of the cellular machinery, which provide a de facto for all the virus needs to generate a new progeny of infectious virions. Most importantly, the processing of the glycoprotein occurs at the _682_RRAR_685_↓ motif and it is mediated by Furin (Figure 1) [15,16,17]. The maturation of S into S1/S2 is crucial for infectivity [15], thus processing represents a major drug target to contain infection [18]. Furin is a member of the basic Proprotein Convertases (PCs) family, whose distinctive mark is the ability of cutting protein substrates after polybasic clusters, typically BX_n_R↓ where B is a basic residue, X is any amino acid except Cys, and n is 0,2,4,6,8 [19]. Proteolytic activation by Furin is likely an important feature that allowed SARS-CoV-2 to overcome the species barriers and thus cause human disease [20]. In the human body, Furin is ubiquitously expressed. Accordingly, any organ becomes a platform for the release of infectious particles, explaining the highly lethal nature and high rate of multiple-system failure caused by SARS-CoV-2 colonization [21]. In contrast, protease cleaving after single basic residues (e.g., airway trypsin-like protease HAT) is typically restricted to specific body parts, such as the upper respiratory tract or intestine [22]. Thus, pathogens depending upon this class of enzymes show more restricted host invasion [23].

The Furin cleavage site (FCS) of the envelope glycoprotein S is the result of a 12-nucleotide insertion encoding for an extra _681_PRRA_684_ motif, switching the solitary arginine **R**↓ into the polybasic cluster P**R**RA**R**↓ [24,25]. This special insertion rendered the glycoprotein sensitive to Furin, nodding to the speculations that the virus may be artificial. Indeed, SARS-CoV-2 S processing is exceptional when compared to other SARS-related coronaviruses. However, sequence analysis has shown that the natural evolution of the S glycoprotein is highly probable [26,27]. A direct consequence of the insertion is the easier maturation of the SARS-CoV-2 spike, which is processed to higher efficiency rates compared to the cognate glycoprotein of the SARS-CoV [28]. The extra PRRA motif was a random event that generated a minimal Furin consensus site within the S polypeptide that started to acquire further mutations as a consequence of the interaction with the enzyme. Due to the large number of infected people worldwide, several variants carrying single or multiple amino acid replacements around the FCS have been reported. Public health agencies—such as the World Health Organization—have assigned some mutants as variants of concern (VOCs). Over the course of the pandemic, two of these VOCs (B.1.1.7 [Alpha] and B.1.617.2 [Delta]) have independently risen to prominence, rapidly replacing the previously circulating strains [29]. The FCS spike from the Alpha and Delta lineages show P681R and P681H replacements, respectively. Most recently, a new isolate has been taking over, namely the Omicron variant. The latter harbors multiple spike mutations, two of which—N679K and P681H—sit close to the Furin scissile bond [30].

Several studies have disclosed that the presence of a Furin cleavage site at the S1/S2 boundary confers additional advantages to the virus, which acquires an increased fitness in terms of an overall enhanced ability to infect [15,31,32,33]. The gain-of-function (GOF) phenotype relies on the ability of the virus to exploit the polybasic cluster for further activities besides using it to mature the envelope glycoprotein. As time goes on, various functions have been gradually discovered, shedding some light on a complex scenario in which RRAR_685_↓ turns out to have prominent roles within the whole virus life-cycle. At the entry level, FCS was found to be a key factor for receptor binding by grossly favoring virus attachment to the main cellular receptor, the angiotensin-converting enzyme 2 (ACE2) [34]. Moreover, FCS broadens the receptor options by mediating the interaction between the envelope glycoprotein S and neuropilin-1 (NRP1) [35] or Heparan Sulfate (HS) [36,37]. NRP1 and HS recognize the C-term Arg and XBBXBX binding motif, respectively, present within the viral _680_SPRRARS_686_ stretch. The FCS was further shown to be required for the following steps of fusion, syncytium formation [15,38] and human cell invasion [39]. Furthermore, the Furin cleavage reduces SARS-CoV-2 sensitivity to innate immune restriction, mediating an early escape from the IFN-induced transmembrane (IFITM) sensors [31]. Vice versa, synthetic variants lacking the polybasic motifs were shown to possess attenuated virulence. Accordingly, despite the appearance of multiple different variants worldwide, natural FCS deletion mutants—e.g., Δ_680_SPRRA_684_ and Δ_685_RSVA_688_—are extremely rare events, suggesting that the cleavage site is under strong selective pressure in humans [40]. Of note, the removal of the entire _679_NSPRRAR_685_ stretch may easily arise when the virus is cultured in vitro [41]. Along these lines, SARS-CoV-2 was shown to rapidly adapt upon culture Vero E6 cells, due to increased cleavage efficiency by cathepsins at the mutated S cleavage site [39]. That is, S1/S2 switching to other proteases is possible, although this option is restricted in vivo, suggesting that there is a tight virus–enzyme interconnection through a mechanism(s) that has yet to be discovered. Finally, it is noteworthy that Furin masters a plethora of functions in virtually all organs. It is conceivable that the massive production of viral spikes competes with endogenous substrates, leading to enzyme dysregulations to further favor viral infection. An example is offered by the sodium epithelium channels (ENaCs), protein complexes critical for the homeostasis of airway surface liquid, whose misregulation is associated with adverse respiratory conditions. ENaC is activated by Furin but undergoes alterations when the envelope glycoprotein S of SARS-CoV-2 is expressed [42].

Understanding the impact of cleavage-site mutations on spike cleavability is important for fighting against SARS-CoV-2 and designing new antivirals. Whether a single amino acid replacement enhances maturation may explain why some viral strains are widespread. Additionally, it may provide evidence that we should be on the alert for forthcoming variants outbreaks. In parallel, new residues at the cleavage site may provide new targetable interactions to develop potent drugs. Importantly, the most-spread variants represent unique events to best pinpoint Furin substrates. The enforced co-existence between SARS-CoV-2 and Furin and the intrinsic virus propensity to mutation resulted de facto in a large in vivo library screening. Indeed, a huge number of combinations of amino acids around the cleavage site has entered into contact with the enzyme, with those with enhanced viral fitness being more successful.

In this study, we focused on the Furin cleavage site of the SARS-CoV-2 glycoprotein S by exploring the GISAID sequences database and reproducing in vitro the most significant mutations to understand the mechanism(s) of adaptation of the pathogen to the human Furin protease. More in detail, we analyzed the SARS-CoV-2 spike sequences of circulating strains and found that major mutations occur upstream of the cleavage site, whereas RRAR_685_↓ is highly conserved. In particular, P681 is either replaced by Arg (Delta variant) or His (Beta/ Omicron variant), but not Lys, despite our studies on synthetic substrates suggesting that it may represent a suitable alternative. Distant variants—Q675H, Q677H and T678I—confer an increased cleavability to 17mer substrates mimicking the cleavage site. Of note, the presence of a histidine is a mark for a broader pH optimum. Peptide conformational investigations exclude major structural differences among mutants. Thus, the increased cleavability points the finger towards an improved substrate–enzyme interaction. Taken together, our findings refine our current knowledge on Furin substrate specificity. In particular, we now understand the importance of the amino acids neighboring the minimal cleavage site RXXR↓.

## 2. Results

### 2.1. SARS-CoV-2 Spike S Varies Close to the Furin Cleavage Site Preferentially at P Positions 

The way a substrate is recognized by Furin is not symmetrical since six amino acids before and only two after the scissile bond—P6-P2′ positions—match the catalytic pocket [43,44]. While scientists agree that polybasic amino acids are a “must” at P positions, little is known on the preference of Furin within the P’ stretch. The MERPOS database (https://www.ebi.ac.uk/merops/index.shtml, accessed on 1 March 2022) shows that Ser and Val are favored at P1′ and P2′, respectively, within a library of 90 officially recognized Furin substrates. Flanking residues, both upstream of the P6 and downstream of the P2′ positions, can still be important in terms of making the core available to the enzyme [45]. Thus, the latter can modulate the kinetics of cleavage, rendering a protein a superior or lousy substrate.

Based on these premises, at the level of the SARS-CoV-2 spike maturation site, we expected little or no variations of the key Arg at P1 and P4, as well as Ser and Val at P1′ and P2′, respectively. In contrast, we anticipated variations of the surrounding residues to be more likely. The SARS-CoV-2 glycoprotein S contains a unique extra “PRRA” motif at the S1/S2 boundary (P2-P5 positions) that is not present in the other coronaviruses such as SARS-CoV-1 and MERS-CoV [24,46] (Figure 1).

The “appearance” of this polybasic cluster changed the consensus motif of the envelope glycoprotein, and the coronavirus became Furin-dependent, in spite of a non-ideal distribution of amino acids all around it. It is conceivable that during the massive viral spread among the human population, the selected random mutations occurring within the spike glycoprotein may be associated to an enhancement of cleavage by the Furin enzyme [47]. To verify this hypothesis, we first performed virtual amino acid scanning by replacing each residue within the 671-700 region (P15-P15′) of S wild-type (Reference sequence _671_CASYQTQTNSPRRAR↓SVASQSIIAYTMSLG_700_; isolate Wuhan-Hu-1; GenBank: MN908947.3) with the other 19 available alternatives and searched for actual variants based on the sequences deposited in the GISAID database (https://outbreak.info/situation-reports, 30 August 2022) [48] (Figure 2 and Appendix A). More in detail, we searched for the existence of SARS-CoV-2 spike variants carrying one single mutation at a time. All possible combinations were explored, e.g., C671 was replaced by any residue among R, K, H, D, E, Q, N, T, S, F, Y, W, P, I, L, V, G, A, and M. Along the same lines, all amino acids surrounding the cleavage site (P15-P15′) underwent a similar analysis. The number of glycoprotein sequences carrying a specific mutation and deposited in the database was annotated accordingly (Appendix A) and graphically represented in a 3D plot. Graph spikes highlight major mutations that occurred and the occurrence of each one of them (Figure 2). The analysis was performed on 30 August 2022.

Overall, we found that residues at P positions vary much more than those at P’ positions, in line with the evidence that the Furin catalytic site preferentially recognizes stretches before the scissile bond rather than after. Thus, the refinement of the trait upstream of the cleavage overtakes that of the downstream part. More in detail:The replacement of the key R685 and R682 at P4 and P1, as well as S686 at P1′, is a rare event (1108, 627, and 193 over 12.335.229 total sequences, respectively), as expected. Interestingly, we performed a similar database analysis on 10 January 2022, thus excluding the SARS-CoV-2 sequences uploaded in the GISAID platform from 11 January to 30 August 2022. Comparing the two analyses, we found that mutations at R682 are recent acquisitions. Indeed, more than 90% of the actual variants have been reported this year (52 vs. 575 from 11 January to 30 August 2022);The most popular variations occur at odd positions (P5, P7, P9, and P11) by replacing WT residues with basic amino acids. Of note, the closer the position is to the cleavage bond, the higher the number of recorded variants.Position P1, cleavage site, R_685_↓Position P5, P681H + P681R ---10.861.932 mutantsPosition P7, N679K -------------- 5.217.144 mutantsPosition P9, Q677H -------------- 83.139 mutantsPosition P11, Q675H ------------ 44.137 mutantsThe variation of residues at even-numbered positions (P6, P8, and P10) is much less frequent. Here, mutations involve the accommodation of amino acids with a hydrophobic character;Finally, there is nothing noteworthy within the P’ region, except the conservative A688V mutation at P3′.

Grouping the mutants by position, we observed a clear cluster of hydrophilic variants increasing around the region before the cleavage site, whereas an opposite trend was found for the stretch following the scissile bond. Interestingly, some classes of amino acids are under-represented, such as the acidic D and E (Figure 3).

Eventually, for each position within the P15-P15′area, we scored the most frequent amino acid replacements, taking them individually (Figure 4).

C671 was no match for any other amino acid, underlying its prominent function. In fact, Cys at 671 is engaged in a disulfide bridge with Cys 662 and it is crucial for the overall architecture of the spike protein [49]. Similarly, Tyr 695 is also invariable, probably due to the very favorable T-shaped p stacking interaction established with the side chain of the aromatic Tyr 660, stabilizing the protein structure. Actually, Tyr 695 and Tyr 660, together with Ile 693 and Val 656, form a singular highly hydrophobic pocket (Figure 5).

The P14-P11 tetrapeptide accommodates conserved replacements: hydrophobic with hydrophobic, aromatic with aromatic, and hydrophilic with hydrophilic. P10-P11 are promiscuous. Interestingly, the P9-P3 stretch—harboring the PRR motif—has the propensity to switch its physical chemical properties. That is, the amino acids corresponding to the insertion have the tendency to acquire a different identity when compared to the original circulating strain. Of note, we observed the rise of Trp-containing variants, though this happens in a limited number of sequences. Tryptophan possesses a bulky aromatic side chain and therefore it may impose local structural constraints to the substrate, right into the catalytic pocket. With regards to the P1′-P15′ positions, the majority varies in a conservative fashion (Figure 4). Overall, we verified that the key residues required for recognition by the Furin enzyme are tightly preserved. Variations of the S1/S2 boundary occur preferentially at P positions, with the appearance of selected amino acids at odd-numbered positions.

### 2.2. In Vitro Digestion of Peptides Mimicking the Cleavage Site of SARS-CoV-2 Spike Variants

#### 2.2.1. Neutral pH Privileges Cleavage of SARS-CoV-2 S WT

The Furin enzyme has a pH optimum of around 7.0, with more than 50% of its enzymatic activity between pH 5 and 8, depending on the substrate being cleaved [50,51]. The broad pH spectrum reflects the ability of Furin to work in different subcellular compartments, including Golgi, endosomes, and plasma membranes [52], each of these having unique [H^+^] contents. Indeed, in eukaryotic cells, the steady-state pH of intracellular compartments varies greatly. Typically, organelles along the secretory pathway are characterized by progressive acidification, from the pH 7.2 of Endoplasmic Reticulum to the pH ~6 of late Golgi stacks [53]. Most envelope glycoproteins of Furin-dependent viruses are cleaved in their way out through the secretory pathway, prior to reaching the viral budding platforms. This is the case of influenza [54], paramyxo [55], retro [56], and hemorrhagic fever [57] viruses, among others. Similarly, SARS-CoV-2 S was found to be cleaved by Furin in the TGN [17]. To investigate the cleavability of SARS-CoV-2 S by Furin in different pH conditions, we transiently expressed the viral glycoprotein lacking the transmembrane domain and the C-term His tag [4] in HEK293 cells. Supernatants were collected at 72 h post transfection (p.t.) and purified by ion-exchange chromatography. Of note, the purified protein contained a fraction of cleaved S, the latter generated by endogenous Furin. Digestions were carried out at 37 °C with soluble human Furin (sFur) [58] at pH, 5.5, 7.0, and 8.0 (25 mM sodium acetate pH 5,5 or 25 mM HEPES (4-(2-hydroxyethyl)-1-piperazineethanesulfonic acid) pH 7.0, supplemented with 2mM CaCl_2_). Relative sFur activity was validated on the Pyr-RTKR-AMC substrate and Furin-specific inhibitor Dec-RVKR-chloromethylketone (Appendix A). Samples were collected at 1, 4, and 7 h and O/N. For each time point, the cleavage reaction was promptly stopped by adding EDTA 0,5M prior to boiling at 95 °C for 5 min. All samples were then stored at −80 °C until final analysis by Western blotting (WB). We found that all three pH levels tested (5.5, 7.0, and 8.0) were suitable for inducing—although differently—full protein processing (the disappearance of S O/N). Accordingly, basic and neutral conditions were more favorable, in particular at early time points (Figure 6a). Interestingly, the conversion of S to S1/S2 was very fast upon the addition of the enzyme (compare control vs. t_0_, (Figure 6a)) and slowed down thereafter, possibly underlying some regulatory mechanisms of Furin activity when products are generated. We repeated an analogous experiment using the synthetic _673_SYQTQTNSPRRAR↓SVAS_689_ stretch as a substrate. The sequence encompasses the cleavage site (P13-P4′) of the envelope glycoprotein S. Briefly, the 17mer peptides were chemically synthesized (Solid Peptide Phase Synthesis (SPPS), Fmoc-chemistry), purified and characterized following standard protocols. The peptide was incubated at 5 μM with recombinant soluble human Furin in suitable buffer (pH 5.5, 7.0, and 8.0) at 37 °C. Digestions were performed as independent triplicates. Samples were collected at regular intervals over 6 h and processed as described above, with the final analysis including reverse phase (RP) high-performance liquid chromatography (HPLC) coupled with ultraviolet (UV) detection (214 nm) to visualize the area corresponding to the unprocessed peptide (Appendix A). Areas relative to the intact peptide were calculated and plotted over time to visualize the disappearance of the substrate following cleavage by Furin. Cleavage at R685 was confirmed by mass spectrometry analysis (Appendix A). In these settings, pH 7.0 was by far the most efficient condition to promote cleavage (Figure 6b and Appendix A).

Overall, the data are compatible with a processing event occurring in the Golgi, as previously reported [17]. In this compartment, the SARS-CoV-2 glycoprotein passes through to meet the TGN-resident Furin enzyme prior to reaching the plasma membrane. For the peptide, the processing pH optimum is centered around neutral values, although this is less striking when considering the entire S molecule. Possibly, regions outside the cleavage site are able to influence the interaction between the protease and its substrates. This finding is consistent with other reports where single-point mutations close to the scissile bond were found to impact cleavage differently, either according to the actual S backbone where they are inserted and/or based on the substrate length. Kinetics further suggest that early times are crucial to generate the mature forms.

#### 2.2.2. Histidines Upstream of the Cleavage Site Result in Gain-of-Function (GOF) Variants

Histidines are the most represented residues across all mutations (Figure 2). These are “special” amino acids, being protonated in a slightly acidic environment or bearing no charge at pH ≥ 7. Indeed, at low pH, both δ-nitrogen and ε-nitrogen of the imidazole ring of the side chain are protonated and the amino acid shows an overall +1 charge. At high pH, the histidine is neutral, with neither δ-nitrogen nor ε-nitrogen being protonated. Due to their bivalence, histidines are known to be master regulators of several biological functions. The rate of conversion of a given substrate into its cleaved products is no exception, e.g., the PrM of Dengue virus contains a key H68 close to the processing site, which is crucial for maturation [49]. Therefore, it is conceivable that the consistent appearance of SARS-CoV-2 variants carrying His residues around the cleavage site of the glycoprotein S is an indication of the adaptation of the virus to the host, in order to better exploit the Furin protease at broader pH conditions. This can be achieved by increasing the substrate/catalytic pocket affinity and/or by the adoption of a more favorable conformation so that the cleavage stretch may dock the enzyme easier. To understand the impact of histidines on the processing event, we approached the question by studying the cleavability of synthetic 17mer peptides encompassing the SARS-CoV-2 spike (P14-P4′) region (Table 1). We focused our attention on P681H (Alpha), Q677H, Q675H and the recently identified N679K/P681H (Omicron) variants. 

P681H was first identified in the UK in late 2020 [59] and then classified as the VOC variant “Alpha” [60]. It was proposed to increase the accessibility of the cleavage site, leading to enhanced processing by a mechanism that relies on the loss of a nearby O-glycosylation that is P681 dependent [27]. The same mutation may also impact the local conformation and thus further favor the binding affinity of the S protein to Furin [61]. The increase of S maturation into S1/S2 is controversial [62,63,64]. Very recently, a fluorogenic 11mer peptide (P8-P3′) carrying P681H was shown to be digested similarly to the analogous WT peptide at pH 7.5 and poorly in acidic conditions [64]. Q675H mutation arose independently in separate evolutionary clades in early pandemic times, and it was suggested to favor processing by conferring a lower structural variability to the Furin cleavage-site loop [65]. Q677H also emerged early in multiple lineages of the SARS-CoV-2 spike protein [66]. Both Q675H and Q677H are VOCs and have been linked to an increased SARS-CoV-2 fitness [67]. Finally, the N679K/P681H Omicron variant is the currently circulating strain, detected for the first time at the end of 2021 and carrying an unusual cluster of mutations, especially within the envelope glycoprotein [68]. Intriguingly, while the full-length S was shown to be cleaved less efficiently than the WT [69], the fluorogenic peptide _678_TKSHRRAR↓SVA_688_ was shown to be a superior Furin substrate [70]. The digestions and analyses of our 17mer peptides were carried out as described above, monitoring the cleavage extent after 1 hr of incubation at 37 °C. Data were corrected for HPLC injection, taking as reference an invariable impurity detected at ~ 21 min (Appendix A). Due to the sensitivity of histidine to acidic conditions, reactions were performed at pH 5.5, 6.5 and 7.0. We observed that all peptides were cleaved better than the WT in acidic conditions (Figure 7a–d,g). Of note, digestions at pH 5.5 seem to occur very quickly, with the precursor at t_0_ already being partially cleaved. Therefore, peptides appear to perform similarly to the full-length S that drastically decreased upon Furin addition (Figure 6a). The Omicron peptide carrying the double N679K/P681H mutation is a superior Furin substrate at all pH values tested, in line with previous reports [70]. Q677H is the worse replacement, being the corresponding peptide poorly digested at neutral pH and to some extent at pH 5.5. Interestingly, the latter shows a profile similar to that found for the P681R peptide (Figure 7f), which mimics the cleavage site of the Delta variant, one of the very early widespread mutants [71]. The poor cleavability of the P681R peptide was quite unexpected since very short sequences from Delta SARS-CoV-2 are reported to be GOF variants [72]. Discrepancies may be due to the different lengths of the peptides used in this work. Moreover, unlike [72], we strictly opted for natural amino acids without the use of additional groups that may favor cleavage readout (e.g., fluorescent groups) but—at the same time—may alter kinetics. In line with these thoughts, it was suggested that P681R contributes to an increased spike cleavability, although this mutation is not fully responsible [62]. Further, Zhang and colleagues showed that P681R affects full-length spike processing by modulating its glycosylation. Thus, the abrogation of specific glycosylation pathways makes P681R irrelevant to the cleavage extent [73].

In summary, the replacement of WT residues with histidines near the scissile bond renders the peptide more cleavable in acidic conditions. Thus, P681H, Q675H and N679K/P681H overcome the WT peptide processing outcome. The ability of single mutations beyond the minimal RRAR_685_↓ motif to modulate Furin activity is astonishing in these examples. Targeting this peripheral region may be a preferential approach to blocking glycoprotein maturation, as an alternative to directly attacking the polybasic motif.

#### 2.2.3. Why Are There No P681K Variants of the SARS-CoV-2 Spike?

Position 681 of the envelope glycoprotein S of SARS-CoV-2 is being kept under monitoring since variants carrying mutations at this position have been linked to VOC forms of the virus. Originally, the envelope glycoprotein accommodated a proline residue at this position. Among Furin substrates, prolines are not very popular at the cleavage site because of their intrinsic rigid structure that confers restricted grades of freedom to the characteristic five-atom ring of proline. Accordingly, it is generally accepted that the substrates of Furin are characterized by a vast degree of flexibility, to allow optimal fitting into the catalytic site [74]. In this respect, it is not surprising that the proline at the 681 position is subject to replacements. Moreover, the mutations at this particular position further make the stretch more accessible since the close-by P681-dependent (O-)glycosylation, which may hamper the substrate–enzyme docking, is lost [53]. Overall, there is a clear pressure on the 681 position for the acquisition of a more suitable residue. P681R (Beta) and P681H (Delta/Omicron) are the most predominant over all possible amino acidic alternatives available (4407836 R; 6454096 H; 0 K; 2 D; 0 E; 8 Q; 36 N; 7 T; 129 S; 4 F; 826 Y; 0 W; 1685 L; 0 V; 0 I; 2 A; 28 C; 0 M; 10864663 TOT sequences deposited in GISAID database as of 30/08/2022). Intriguingly, no lysine has ever been reported to replace proline at 681, despite Arg, His and Lys sharing basic features. Indeed, the side chains of the three residues are positively charged. We reasoned that Lys may not optimally fit the catalytic pocket of Furin due to its side chain, which may result in it being too long. To test this hypothesis, we synthesized two 17mer peptides encompassing the SARS-CoV-2 spike (P14-P4′) region and bearing either the P681K or P681Orn mutation (Table 1). Ornithine is not an amino acid coded by DNA. Despite not being a building block of proteins, the residue is an important natural intermediate in metabolic processes of mammalian tissues. The ornithine side chain (-[CH_2_]_3_-NH_3_^+^) closely resembles that of lysine (-[CH_2_]_4_-NH_3_^+^), but it lacks a single -CH_2_ unit. That is, Orn is a “short version” of Lys. Peptides were digested and analyzed as reported above. We found that the P681Orn peptide behaved similarly to the WT at pH 7.0 and did not acquire cleavability in acidic conditions (Figure 7a vs. Figure 7e). On the other hand, P681K acquired cleavability at acidic pH values when compared to the WT (Figure 7a vs. Figure 7h). Overall, P681K performance was better than that of P681Orn and WT itself.

In conclusion, the presence of lysine instead of proline at position 681 may be beneficial in terms of peptide cleavability that becomes sensitive to mild acidic conditions. The length of the side chain of Lys may be crucial, given that the shorter ornithine analogue reverts the Lys phenotype. Overall, SARS-CoV-2 provides new insights on Furin specificity, also through the negative in vivo selection of specific mutations.

#### 2.2.4. Non-Conservative T678I Replacement Enhances Processing

The increase in basic residues around the scissile bond is compatible with the heavily positively charged surface of the catalytic pocket of Furin [75]. Within the SARS-CoV-2 spike, Arg/Lys/His replacements are preferentially found at odd positions and upstream of the main RRAR_685_↓ motif. In contrast, hydrophobic residues are more representative for substitutions at the even positions (Figure 5). T678I replacement was detected early in 2020 [76] and it now represents one of the most frequent mutations after the Beta/Delta and Omicron variants (Appendix A). To our knowledge, no information is available regarding the impact of T678I on glycoprotein cleavage. To address this question, we synthesized the corresponding 17mer peptide (Table 1) that underwent digestion, as reported above. The substrate showed a two-fold cleavability increase with respect to the wild-type version at pH 7.0, and also some sensitivity to processing in acidic conditions (Figure 7a vs. Figure 7i).

Among all variants tested here, T678I stands as one of the best GOF mutants of the SARS-CoV-2 glycoprotein. Importantly, the data further suggest that combinations of specific amino acids can modulate the activity of Furin. More in detail, hydrophobicity—contrary to the expectations—can be a “plus” for substrate processing if strategically located upstream of the RX_n_R↓ sequence.

### 2.3. Conformational Investigations of the SARS-CoV-2 Wild-Type Cleavage Site and Its Mutants

Single mutations around the conserved RRAR↓ motif do greatly influence the processing event. In order to verify whether this phenomenon could be attributable to any conformational change, we performed Circular Dichroism (CD) and Attenuated Total Reflectance (ATR) Fourier transform–infrared spectroscopy (FT-IR) studies on the various peptides synthesized in our study (Figure 8 and Figure 9). CD investigations in water suggest that the entire panel of sequences adopt no special conformation, demonstrating typical random CD profiles (Figure 8a). Accordingly, disordered proteins possess very low ellipticity above 210 nm and negative bands near 195 nm [77]. The flexibility of peptides is further evident when the latter are dissolved in trifluoroethanol (TFE), a solvent known to prompt helical structures. Here, we do observe the appearance of the characteristic α- helix positive values below 200 nm and the negative peaks around 222 and 208 nm [78]. Peptides possessing mutations at 681 position are slightly red-shifted (Figure 8b).

CD analyses were complemented by ATR-FTIR investigations using lyophilized peptides. Spectra were collected in the 400–4000 cm^−1^ range (Figure 9a). We focused our attention on amide I and II bands, which are the most characteristic absorptions in protein spectra (Figure 9b–g). Amide I is the result of the stretching vibrations of C=O and C-N groups. Its frequency is found in the range between 1600 and 1700 cm^−1^. Amide II derives mainly from in-plane N-H bending and C-N/C-C stretching vibrations. It is found in the 1510 and 1580 cm^−1^ regions. Overall, the exact band positions are due to the backbone conformation and the hydrogen bonding pattern (typically, 1623–1641 cm^−1^ β-sheet; 1642–1657 cm^−1^ random coil; 1648–1657 cm^−1^ α-helix; 1674–1695 cm^−1^ β sheet) [79]. We found that His containing peptides N677K/P681H, P681H and Q675H, but not Q677H, show and additional peak around 1618 cm^−1^. N677K/P681H and Q675H peptides are further characterized by an increased 1618/1650 cm^−1^ ratio (Figure 9b). No clear shifts were detected in the 1480–1580 cm^−1^ range (Figure 9c). When clustering the peptides according to mutation at position 681, we could observe that the replacement of WT proline with a basic residue (Lys, Arg, and Orn) did not grossly affect the ATR-FTIR profile in the 1580–1690 cm^−1^ range (Figure 9d). However, the same mutations induced a variation within the 1480–1580 cm^−1^ range, resulting in a major contribution of the band component, peaking at around 1537/40 cm^−1^ (Figure 9e). No differences were evident in the case of the T678I peptide (Figure 9f,g).

Overall, the data at hand suggest that variants may acquire a certain degree of order when compared to the WT peptide sequence.

### 2.4. Sustainability

We are deeply engaged in sustainability. Being aware of the resources employed in our work is of great importance to better plan future experiments.

The entire project was monitored for the use of disposable plastics (Table 2). In parallel, ad hoc measures were adopted to minimize the impact of our work on the environment and non-renewable resources, whenever it was possible.

According to our “Green Book” (GB) summary, we required approximately 10 kg of plastics that were disposed as recyclable resources only in part (10%). The rest was stocked as hazardous materials for the following treatment, as stated by Italian laws. During the realization of this study, we replaced some plastics with equivalent glass items, focusing on those steps where it was more necessary. For example, we opted for re-usable glass vials to collect purified peptides and monitor digestions by HPLC. Despite one single HPLC vial weighing only 0.1 g and one single collecting tube weighing only a few grams, since we ran hundreds of independent analyses, it is clear that we saved considerable plastic waste.

We believe that researchers must become more aware of the disposable items that are used in their laboratory on a routine basis. A simple diary (e.g., GB) reporting a rough estimation of plastics use may be useful for visualizing possible weaknesses, allowing researchers to intervene with greener alternatives.

## 3. Discussion

SARS-CoV-2 is the etiological agent of COVID-19, the current pandemic claiming millions of lives. Due to the SARS-CoV-2 spread worldwide and its massive propagation, the pathogen has the chance to grossly mutate and eventually adapt to the human host. Following the initial waves of infection, the pathogen has shifted towards variants with adaptive advantage over previous strains [8,9,10,80]. In this work, we have focused our attention on specific mutations located at the S1–S2 boundary of the envelope glycoprotein S.

To be functional, the virus requires the maturation of the glycoprotein S by cleavage at the _682_RRAR↓S_686_ motif by Furin. The presence of a multi-basic site distinguishes SARS-CoV-2 from SARS-CoV and all other known sarbecoviruses whose S protein is not cleaved by Furin-like proteases [80,81]. The processing was shown to decrease the overall stability of the SARS-CoV-2 glycoprotein when compared to the S protein of a closely related bat virus (RaTG13) lacking the Furin cleavage site (CS). In turn, this implied the easier adoption of the open conformation required for SARS-CoV-2 S to bind to the human ACE2 receptor. Indeed, RaTG13 virus would not be able to interact with the human receptor efficiently and would thus be unlikely to infect humans directly [4]. Another important aspect of the CS is its dependence on Furin—a ubiquitous protease –which enables SARS-CoV-2 to replicate in virtually any organ, allowing massive human body infection. Importantly, the virus has not been reported to escape from Furin in vivo, despite this being shown to be readily feasible in vitro [40].

Based on these premises, the appearance of the Furin cleavage site within the SARS-CoV-2 glycoprotein has been a key event that has drastically shaped its properties. Of course, strictly monitoring likely modifications around the cleavage site is of great importance because any variation may have a significant impact on virus infectivity and tropism. As for other viral sequences, the region around the processing site has evolved and acquired multiple mutations over time. By analyzing the GISAID sequence database, we could observe that the polybasic cluster is rather conserved. Our report is perfectly in line with the evidence that SARS-CoV-2 continues to prefer Furin, a protease that cleaves substrates after BX_n_B↓ (B, basic residue; X, any residue but Cys). Despite the phenomenon having no clear explanation, it somehow resembles the case of the Ebola virus (EBOV) glycoprotein, which is cleaved by Furin as well. Ebola virus glycoprotein cleavage by Furin is not critical for virus replication in vitro and infectivity and virulence in non-human primates [82], and the polybasic motif is highly conserved among different strains and isolates. In contrast to the conservation of the motif _682_RRAR↓S_686_, surrounding residues are much more likely to vary. In particular, replacements with hydrophilic amino acids are preferred at P positions and hydrophobic amino acids are preferred at P’ positions. The exact pattern of mutations, as well as the identity of the residues, seems to be everything but random, e.g., position 681 accommodates His or Arg but never Lys, despite the latter sharing similar basic features when compared to the first two. In line with the evidence reported by other groups, we found that many peptides mimicking the processing site of variants possess a gain-of-function phenotype in terms of cleavability. Likely, the mutants do better match their partner, that is, the Furin enzyme. Of note, the Omicron N679K, P681H peptide was shown to be the most cleavable substrate among those tested here. Interestingly, position 681 of the S glycoprotein is five residues far from the scissile bond (P5 position). Accordingly, substrates carrying basic amino acids at the P5 position were recently shown to outperform any other alternative when interacting with the Furin catalytic pocket [83]. The contribution of K679 cannot be underestimated, either. On the other hand, we showed that the P681K peptide possesses a gain-of-function phenotype, although this mutation has never been reported in the GISAID database. The reason why certain amino acids—e.g., the in vitro gain-of-function K681—are not viable among the circulating SARS-CoV-2 variants remains elusive. This suggests that the amino acid at this peculiar position may be involved in functions other than modulating the propensity to cleavage by Furin. Accordingly, the cleavage site functions as a binding motif to alternative receptors [80]. In addition, we cannot exclude that the presence of Lys at position 681 may be detrimental for the overall glycoprotein arrangement, which would lose the optimal affinity for the major receptor: the ACE2 protein [4,80,84]. Finally, there is a mutation—T678I—placed very far away from the actual processing site (R_685_↓) that is able to greatly affect cleavability. It is noteworthy to highlight the hydrophobic nature of this replacement.

The peculiar position and nature of T678I suggest that the Furin consensus sequence BX_n_B↓ represents a minimal recognition motif where all surrounding regions are able to modulate the extent of processing. This is further supported by the distant Q677H and Q675H mutations that provide key information regarding the way Furin works. Interestingly, Q675H increases the substrate cleavability at acidic conditions, whereas the nearby Q677H does not. Of note, we observed minimal changes on the residues after the cleavage site. We may conclude that P’ positions are not as important as P positions since they did not require any gross refinement, that is, the introduction of amino acids different from those found in the wild-type sequence. From a different point of view, the actual residues may be the best Furin option and any replacement may result in a loss-of-function variant. Along these lines, known Furin substrates privilege hydrophobic amino acids at P1′ and P2′ positions, as is the case of the SARS-CoV-2 spike cleavage site.

The question of whether mutations also impact CS conformation still remains open. We found that stretch 673–689 of the glycoprotein S adopts no specific structure in aqueous environments, both in the case of WT and variant peptides. This is in line with the evidence that this region is not visible in the resolved crystal structure [4], further suggesting a high degree of backbone flexibility. The finding is not surprising since the region around the scissile bond is normally thought to be able to match the enzyme by adapting to the rigid structure of the catalytic pocket. However, we observed that some mutations—in particular those introducing a histidine residue—induce a shift in the recorded ATR-FTIR and CD spectra. Despite the variation not speaking for a clear-cut conformational change, the data at hand suggest that the population of conformers may be different in the case of the mutations, and in turn this could impact catalytic pocket/interaction. Interestingly, recent studies on Furin revealed that the enzyme can shape multiple conformations. Each single state does not exist per se, but it is generated by the interactions with specific substrates/inhibitors [66]. Thus, the way each peptide is recognized by the enzyme may be different, and the ultimate structure of the peptide bound to Furin CS may be unique and not predictable by simple modelling. However, the picture is much more complex, and the data collected here are insufficient to fully explain the increase in specific combinations of amino acids but not others. In fact, we need to keep in mind that the glycoprotein S has multiple functions and the stretch encompassing the cleavage site may be involved in activities unrelated to spike maturation. Finally, we cannot exclude that the introduction of amino acids—other than those actually found—at the cleavage site may be well tolerated in terms of Furin cleavability but may likewise create new unwanted functions that are detrimental for virus propagation.

## 4. Materials and Methods

### 4.1. sFurin and SARS-CoV-2 S Production

sFurin consists of a soluble form of hFurin truncated before the transmembrane domain [85], whereas the SARS-CoV-2 S protein lacks the transmembrane region [4]. sFurin and SARS-CoV-2 S proteins were produced in HEK293 F cells (Human Embryonic Kidney cells) and grown in suspension in FreeStyle™ 293 Expression Medium (Gibco, ThermoFisher) in shaking (130 rpm) flasks, without antibiotics, at 37 °C and 8% CO_2_. In detail, cells were sub-cultured every four days to 0.3 × 10^6^ cells/mL density and brought to final 1 L at 1 × 10^6^ cells/mL for transfection by polyethylenimine (PEI). Briefly, for 1 mL of culture, 3 μg of PEI and 1 ug of DNA were resuspended in 20 mL of Opti-MEM (Gibco, ThermoFisher) each, separately. The two solutions were incubated for 15 min at room temperature (RT) prior to mixing together. Following a further 15 min, the final solution was added drop-wise to the cell culture under stirring. Cells were incubated at 37 °C, 8% CO_2_ for four days prior to media collection. Cells were removed by centrifugation (6000× *g* for for 20 min at 4 °C (Backman Coulter^®^ Avanti^®^ 25, rotor JLA 9.1000). Supernatants were aliquoted and stored at −80 °C (sFur) or further treated for purification (SARS-CoV-2 spike). SARS-CoV-2 S protein supernatant was filtered (0.45 μm) prior to injection into a Ni Excell column (CV = 3mL) pre-equilibrated in buffer A (25 mM Tris pH 8, 150 mM NaCl, 10 mM Imidazole). Protein elution was achieved by treating with 4mL of 60% buffer B (25 mM Tris pH 8, 150 mM NaCl, 500 mM Imidazole). The purity grade of SARS-CoV-2 S protein was assessed by SDS-PAGE gel and Coomassie stain.

### 4.2. In Vitro SARS-CoV-2 S Digestion

Protein concentration was assessed by nanodrop. Buffers: 20 mM CaCl_2_, 250 mM Sodium acetate, pH 5.5; 20 mM CaCl_2_, 250 mM HEPES (4-(2-hydroxyethil)- 1- piperazineethanesulfonic acid), pH 7.0; and 20 mM CaCl_2_, 250 mM Tris-HCl, pH 8. A supernatant of cells overexpressing sFur was used as the source of enzymes. Then, 1.5 ug of SARS-CoV-2 S glycoprotein was diluted in suitable buffer to a final 150 μL in the presence of 20 μL of sFur and 20 μL of buffers. For each time point, 36 μL of digestion solution was collected, mixed with 4 uL of ethylenediaminetetraacetic acid/ethylene glycol-bis(β-aminoethyl ether)-N,N,N′,N′-tetraacetic acid (EDTA/EGTA) 500 mM, and heated at 95 °C for 5 min. All samples were frozen at −80 °C prior to a final Western blotting analysis.

### 4.3. Western Blot

Protein samples were treated with Laemmli buffer and incubated for 5 min at 95 °C prior to loading to gels. Protein weights were determined using a Prestained Protein MW Marker (Thermo Scientific, Waltham, MA, USA). Electrophoresis was performed in SurePAGE^TM^ Bis-Tris (4–20%) precast gels (GenScript Biotech, Rijswijk, Netherlands) and Tris-MOPS buffer (pH 7.5). Transfer was carried out in a semi-dry condition from SDS-PAGE to a nitrocellulose membrane, using a Bio-Rad Trans-Blot Turbo Transfer System (90 V for 90 min). Transferred proteins were detected with an anti-His HRP conjugation (Abcam, Cambridge, UK 1:4000). Images were processed with Fiji (ImageJ, https://imagej.net/).

### 4.4. Peptide Synthesis

All peptides were synthesized by solid-phase peptide synthesis (SPPS) techniques (Syro I, Multisynthec, Witten, Germany) using Fmoc-chemistry. Amino acid side-chain protections were: t-Butyl—(tBu)- for Ser, Thr, and Tyr; trityl (Trt)- for Asn and Gln; 2,2,4,6,7-pentamethyIdlhydrobenzofuran-5-sulfonyl group (Pbf) for Arg. As solid support, the rink amide resin (0.52 mmol/g; scale 0.1 mmol) was used. The cleavage of the newly synthesized peptides from the resin and the side-chain global deprotection was achieved by treatment with 5 mL of 2.5% H_2_O MilliQ, 2.5% Triethylsilane (TES), and 95% Trifluoroacetic acid (TFA) for 90 min. Crude peptides were filtered out from concentrated TFA solution and precipitated with cold ethyl ether. Approximately 30 mg of each crude peptide was purified by semi-preparative HPLC in the following conditions: column Zorbax 300SB-C18 (5 μm, 300 Å, 9.4 × 250 mm, Agilent, Santa Clara, CA, USA); eluent A, 0.05% TFA in MilliQ water; eluent B, 0.05% TFA in CH_3_CN; gradient, 0% to 40%B in 40 min; flow rate, 4 mL/min; and detection UV absorption at 214nm. Peptide purity grade was assessed by analytical HPLC to be ≥97% (Vydac 218TP C18 (5 μm, 300 Å, 4.6 × 250 mm, Grace, Columbia, SC, USA); injection volume, 20 μL of 1mg/mL peptide; eluent A, 0.05%TFA in H_2_O milliQ; eluent B, 0.05% TFA in CH_3_CN; gradient, 0–20% B in 40 min; and UV detection at 214 nm). The identity of the peptides was assessed by MALDI mass spectrometry (Table 1).

### 4.5. Peptide In Vitro Digestions

Aqueous peptide stocks were prepared at 5 mM. Buffers and sFur, as described in paragraph 4.2., 2 μL of peptide stock solutions, 20 μL of sFur, and 20 μL of suitable buffer were mixed in H_2_O MilliQ up to a 150 μL final volume. Then, 20 μL of samples was collected at the indicated time points and treated as described in paragraph 4.2. All samples were analyzed by HPLC—Vydac Column 218TP C18 (5 μm, 300Å, 4.6 × 250 mm, Grace, Columbia, SC, USA); injection volume, 20 μL reaction mix; eluent A, 0.05%TFA in H_2_O milliQ; eluent B, 0.05% TFA in CH_3_CN; gradient, 0–20% B in 40 min; and UV detection at 214 nm.

### 4.6. Circular Dichroism (CD) Analyses

CD measurements were realized with a spectropolarimeter Jasco J-810 at 25 °C. The CD signal was monitored at 0.2 nm intervals from 260 nm to 195 nm with a scan speed of 100 nm/min in a 0.1 mm quartz path cuvette, registering ten scans. Peptides were dissolved in acidic buffer (25 mM Sodium Acetate pH 5.5, 2 mM CaCl_2_) and 100% Trifluoroethanol (TFE), at a concentration of 50 μM. The data reported are the average of ten scans, subtracting the background obtained by a blank of either buffer only or 100%.

### 4.7. Attenuated Total Reflectance Fourier-Transform Infrared Spectroscopy Investigation

Lyophilized peptides were analyzed with a Jasco FT/IR -4700 instrument equipped with an ATR-PRO ONE diamond accessory. The system resolution was 0.8 cm^−1^. Samples were placed on the cell measurer and scanned from 4000 to 400 cm^−1^. For each sample, 40 scans were run at room temperature.

## 5. Conclusions

In summary, we investigated the cleavage site of the SARS-CoV-2 glycoprotein and found that circulating strains evolved specific mutations over others. In general, hydrophilic replacements are more common among the positions before the scissile bond, whereas hydrophobic amino acids are more representative of the positions after the cleavage site. In order to understand the impact of specific amino acid replacements on cleavage efficiency, peptides encompassing the spike scissile stretch and carrying the most representative variants were synthesized and digested with Furin in vitro. Mutations generally make the peptide easier to cut. The Omicron-derived N679K, P681H double mutant was found to be a superior substrate for Furin, thus supporting the large spread of this variant among the worldwide population. In addition, we provide the first evidence suggesting that the increase in specific mutations but not others is not only due to the selection of better cleavable Furin substrates. Indeed, despite being cleaved in vitro, the P681K variant has never been reported, supporting a more complex role of the cleavage site during infection. Moreover, for the first time, we provide some information on the structure of this specific region, which seems to be random in aqueous solutions. Despite no peculiar structural motifs being present, replacements of the WT residues with other amino acids induce a likely change in the conformer population.

From the point of view of the Furin enzyme, the emergence of variants carrying specific amino acid replacements at the cleavage site provides important information on protease substrate specificity. Indeed, despite the minimal Furin consensus sequence BX_n_B↓ being well accepted, little is known about the surrounding amino acids that may influence Furin’s ability to cleave its substrates. During the pandemic, the SARS-CoV-2 virus has generated a tremendous number of variants, virtually scanning any possible combination of residues at the cleavage site. Those that provided a gain-of-function phenotype—that is, a better cleavability—have been selected. However, as discussed above, we need to keep in mind that some mutations may have never seen the light of the day because of reasons unrelated to Furin processing.

Further investigations are required to understand the importance of these findings in the context of the full-length SARS-CoV-2 glycoprotein. The entire spike often carries multiple mutations that are not limited to the cleavage site. These additional mutations may be critical for understanding the overall impact of the amino acid replacement studied here.

## Figures and Tables

**Figure 1 ijms-24-04791-f001:**
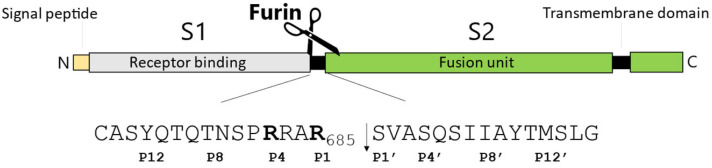
Schematic representation of the envelope glycoprotein S of SARS-CoV-2 and its cleavage site (sequence 671–700). Amino acids, one-letter code. Key arginines for Furin recognition are in bold. The scissile bond is designated by an arrow. Positions upstream (P12 to P1) and downstream of the cleavage (P1′ to P12′) are specified.

**Figure 2 ijms-24-04791-f002:**
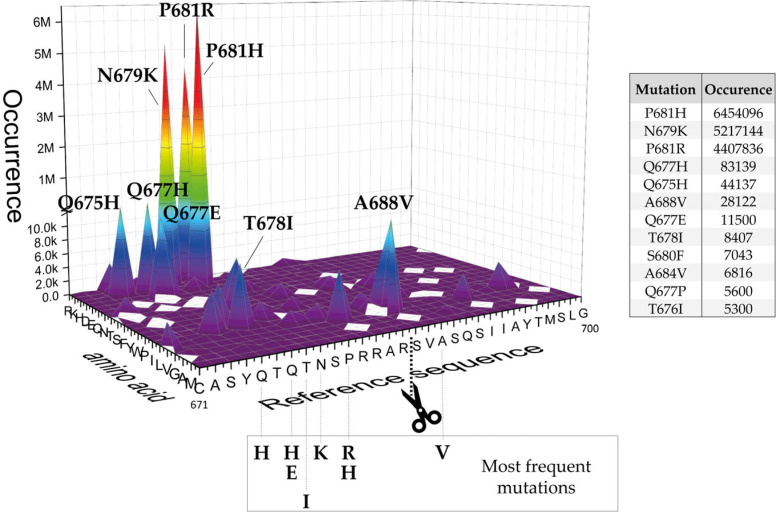
Occurrence of mutants within the SARS-CoV-2 S cleavage site _671_CASYQTQTNSPRRARSVASQSIIAYTMSL_700_ (GISAID sequence database).

**Figure 3 ijms-24-04791-f003:**
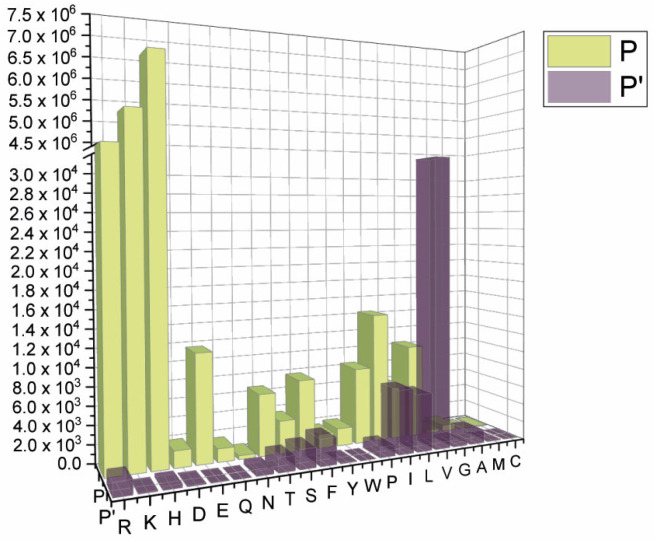
Prevalence of amino acid mutations based on either P (before cleavage) or P’ (after cleavage) positions within the SARS-CoV-2 S cleavage site _670_CASYQTQTNSPRRARSVASQSIIAYTMSL_700_ (GISAID sequence database).

**Figure 4 ijms-24-04791-f004:**
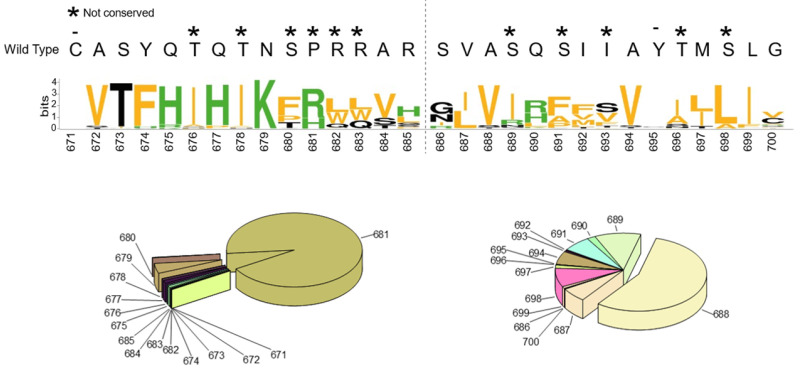
(Top) Sequence logo relative to the spike SARS-CoV-2 S shows the most popular naturally occurrent amino acid replacement for each position, taken individually (https://weblogo.berkeley.edu/logo.cgi, 30 August 2022); (Bottom) Statistic cake images reporting the relative contribution of the sum of all mutant sequences at a given position, grouped according to P (left) and (P’) residues.

**Figure 5 ijms-24-04791-f005:**
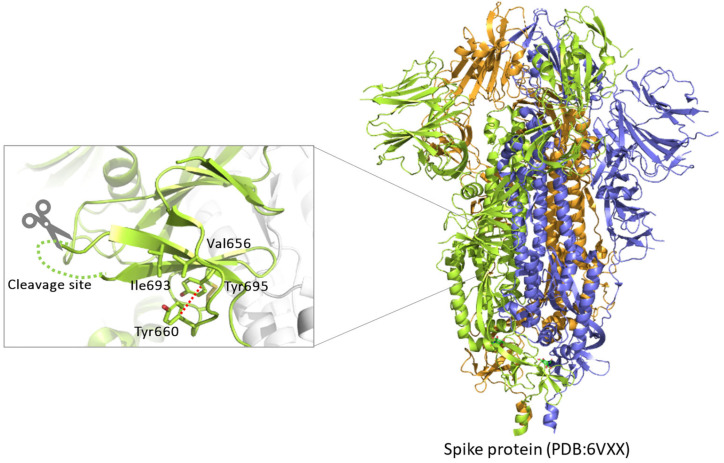
Cartoon view of spike glycoprotein trimers in closed conformation (right panel, PDB 6VXX). Zoomed view (left panel) of one of the cleavage sites: side chains (sticks view) of residues forming an adjacent hydrophobic cleft are evidenced, and the stacking interaction is traced (red dashed line).

**Figure 6 ijms-24-04791-f006:**
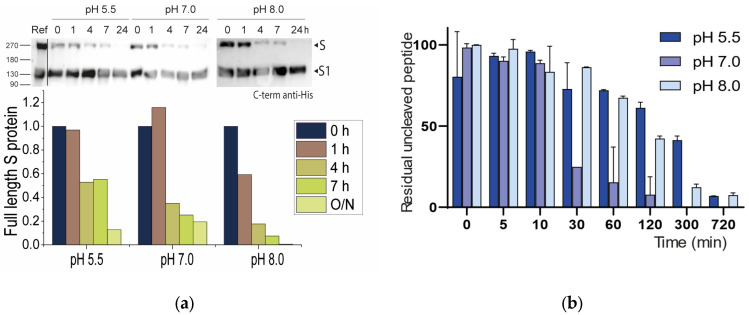
(**a**) Western blot analyses of SARS-CoV-2 glycoprotein cleaved at different pH levels. Bands correspond to the FL protein that disappears upon incubation with sFur over time. Control, FL spike, without the addition of sFur. O/N, overnight. Anti-His antibody was used to detect the protein; (**b**) In vitro cleavage of _673_SYQTQTNSPRRAR↓SVAS_689_. The peptide was incubated with sFur as described in the Materials and Methods section. Extension of cleavage was evaluated as residual uncleaved peptide detectable by HPLC at 214 nm. Areas under the peak of the uncleaved peptide at time 0 min were arbitrarily taken as reference. Experiments were performed as triplicates.

**Figure 7 ijms-24-04791-f007:**
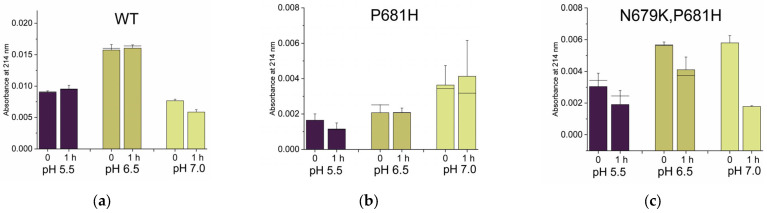
In vitro digestion of peptides mimicking the cleavage site (673–689) of SARS-CoV-2 S WT and mutants. Peptides were incubated with sFur as described in the Materials and Methods section. Areas relative to residual uncleaved peptide detectable by HPLC at 214 nm are reported. Experiments were performed as triplicates. Mean values and error bars are reported; (**a**) WT peptide; (**b**) P681H; (**c**) N679K/P681H; (**d**) Q675H; (**e**) P681Orn; (**f**) P681R; (**g**) Q677H; (**h**) P681K; (**i**) T678I mutant peptides.

**Figure 8 ijms-24-04791-f008:**
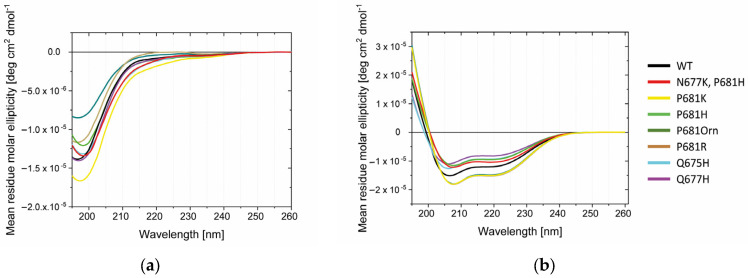
CD spectra of the SARS-CoV-2 S-derived peptides described in Table 1 in (**a**) water and (**b**) trifluoroethanol. Data are expressed as mean residue molar ellipticity (Raw data and calculations are reported in Appendix A).

**Figure 9 ijms-24-04791-f009:**
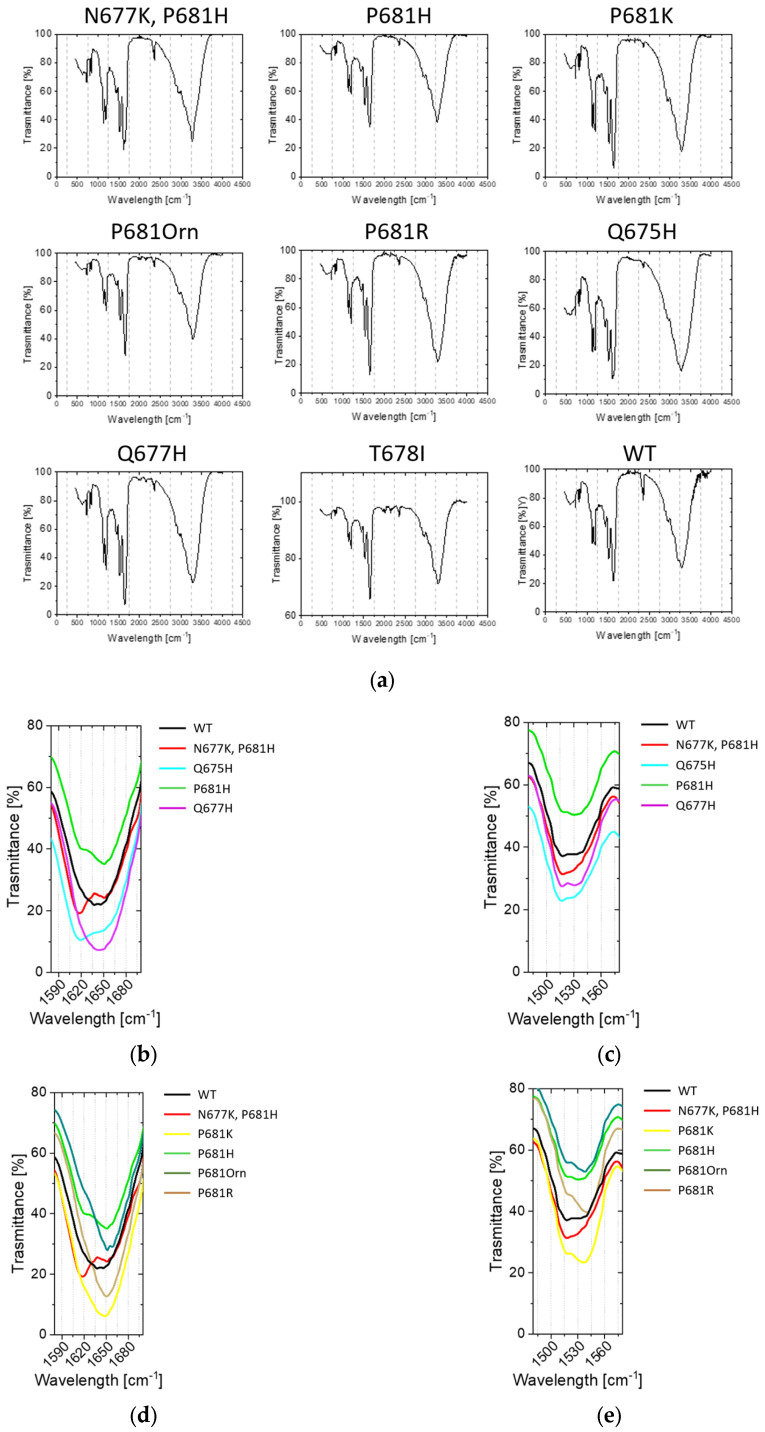
ATR-FTIR spectra of the SARS-CoV-2 S-derived peptides described in Table 1 (**a**) 400–4000 cm^−1^; (**b**,**d**,**f**) 1580–1690 cm^−1^; (**c**,**e**,**g**)1480–1580 cm^−1^. (**b**,**c**) show His mutants vs. WT spectra; (**d**,**e**) show mutations at 681 vs. WT spectra; (**f**,**g**) show T678I vs. WT spectra.

**Table 1 ijms-24-04791-t001:** Sequences (one-letter code for all amino acids except ornithine, which is indicated by Orn) In yellow, mutations vs. wild-type. Theoretical mass refers to the calculated monoisotopic molecular weight; Experimental mass refers to the molecular weight found by MALDI-Mass Spectrometry analyses.

	ID	P13	P12	P11	P10	P9	P8	P7	P6	P5	P4	P3	P2	P1	P1’	P2’	P3’	P4’	Theoretical Mass [Da]	Experimental Mass [Da]
**Naturally occurring**	**Wild Type (WT)**	**S**	**Y**	**Q**	**T**	**Q**	**T**	**N**	**S**	**P**	**R**	**R**	**A**	**R**	**S**	**V**	**A**	**S**	1908.05	1907.61
	**Q675H**	S	Y	H	T	Q	T	N	S	P	R	R	A	R	S	V	A	S	1917.06	1917.69
	**Q677H**	S	Y	Q	T	H	T	N	S	P	R	R	A	R	S	V	A	S	1917.06	1917.52
	**T678I**	S	Y	Q	T	Q	I	N	S	P	R	R	A	R	S	V	A	S	1920.11	1920.63
	**N679K, P681H**	S	Y	Q	T	Q	T	K	S	H	R	R	A	R	S	V	A	S	1962.15	1962.46
	**P681H**	S	Y	Q	T	Q	T	N	S	H	R	R	A	R	S	V	A	S	1948.08	1948.55
	**P681R**	S	Y	Q	T	Q	T	N	S	R	R	R	A	R	S	V	A	S	1967.12	1968.61
**Synthetic**	**P681K**	S	Y	Q	T	Q	T	N	S	K	R	R	A	R	S	V	A	S	1939.11	1939.87
	**P681Orn**	S	Y	Q	T	Q	T	N	S	Orn	R	R	A	R	S	V	A	S	1925.07	1925.65
	**amino acid position**	**673**	**674**	**675**	**676**	**677**	**678**	**679**	**680**	**681**	**682**	**683**	**684**	**685**	**686**	**687**	**688**	**689**		

**Table 2 ijms-24-04791-t002:** Plastic items used in the current work and registered in our Green Book (GB).

Use	Object	Quantity	Unit Weight [g]	TOT Weight [g]
Peptide synthesis	PTFE filter 0.45 μm (diameter 30 mm)	9	2.30	20.70
	PVDF filter 0.45 μm (diameter 13 mm)	9	12.00	108.00
	Reactor	9	4.00	36.00
	Syringe (1 mL)	9	2.20	19.80
	Syringe (20 mL)	12	10.00	120.00
	Tips (200 μL)	100	0.27	27.00
	Tips (20 μL)	100	0.12	12.00
	Tube (1.5 mL)	40	1.00	40.00
	Tube (2 mL)	10	1.15	11.50
	Tube (50 mL)	40	12.70	508.00
	Tube (6 mL)	30	3.50	105.00
	Glove pairs	9	2.30	20.70
In vitro—conformation studies	Flask (125 cm^2^)	1	100.00	100.00
	Flask (250 cm^2^)	1	200.00	200.00
	Flask (75 cm^2^)	6	60.00	360.00
	Plate (24 well)	16	65.40	1046.40
	Plate (48 well)	3	56.00	168.00
	Plate (96 well)	3	64.60	193.80
	Sterile pipette (10 mL)	80	14.00	1120.00
	Sterile pipette (25 mL)	15	15.60	234.00
	Sterile pipette (2 mL)	2	4.40	8.80
	Sterile pipette (5 mL)	60	7.90	474.00
	Tips (10 μL)	100	0.12	12.00
	Tips (1000 μL)	200	0.76	152.00
	Tips (200 μL)	1500	0.27	405.00
	Tube (1.5 mL)	1000	1.00	1000.00
	Tube (15 mL)	15	6.40	96.00
	Tube (50 mL)	30	12.70	381.00
	Glove pairs	200	6.20	1240.00
	Other			1000.00
			**TOT**	**9819**.**00**

## Data Availability

Not applicable.

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
