# Peer review of "SARS-CoV-2 S Mutations: A Lesson from the Viral World to Understand How Human Furin Works"

_ijms, 2023, doi:10.3390/ijms24054791_

Round 1
Reviewer 1 Report
The manuscript, in general, is fascinating. The authors report how punctual changes can modify furin's relationship with the SARs-CoV-2 Spike. Despite exciting data, there needs to be more clarity in the presentation of the data and discussion of the same. Below, I leave some contributions for the authors:
- Line 57 – Remove the information "(Fig 10)"
- Line 175 – 177 – It is necessary to insert a reference that collaborates with this information or reformulate it. Because his study is not about the evolution of SARS-CoV-2 and does not have information that supports the theory of evolution related to the host.
- Line 181 – The need to explain in more detail the information in figure 2 and quote it before its appearance.
- Figure 2 – I believe the image was confusing. It would be more interesting if the alignment pointed out the changes that occurred and the occurrence of each one of them. The unit is missing on the Y-axis.
- Line 190 - Wouldn't it be item 1?
- Line 193 – Lacks a better explanation of this information. I suggest using the graph to explain better.
- Line 195 – Need to demonstrate the veracity of the information regarding the most popular substitutions. A better description of the presented data should help better in this understanding.
- Discussion – I believe this is necessary for further adjustments—mainly the insertion of more information, mainly related to furin and the spike region. An example is article 10.1038/s41594-020-0468-7, where the author has exciting information about furin's function, which is not well discussed in his manuscript.
I suggest that authors read the following papers to enhance the discussion:
10.1038/s41580-021-00418-x
10.1002/jobm.202000537
10.1002/jmv.26615
https://pubmed.ncbi.nlm.nih.gov/32275259/
- Line 557 – Further discussion is necessary for this conclusion to make some sense concerning the data generated in this work.
I am anxious and hopeful that the manuscript will be modified and thus make the results more straightforward.
Reviewer 2 Report
The manuscript by Cassari et al. “SARS-CoV-2 S mutations: a lesson from the viral world to understand how human Furin works” demonstrated the vital experimental details of the development of drugs targeting Furin and Furin-dependent pathogens. Overall, the manuscript is experimentally well demonstrated and requires revision before its publication as follows:
Comments
1. Lines 46-51, the information can be more elaborated as i) variation in human lifestyle like diet along with microbiota that plays a crucial role in immunity and health against viral infections, and severity of viral infection especially SARS-CoV-2 to their constant evolution (mutation/selection) via shifting host from animal to human infection, i.e., development of various SARS-CoV-2 variants and iii) challenges of SARS-CoV-2 superior mutants (novel) on the global pandemic in the near future i.e., Sig. Transduct. Target. Ther. 6 (2021) 226; Infection 50 (2022) 309–325; Microbial Pathogenesis 173 Part A (2022) 105828; Indian Journal of Microbiology 60 (2020) 420–429.
2. Please provide one illustrations-based on the results finding to highlight the significance of this study.
3. Discussion is weak, more significance of finding based on quantitative data can be discussed.
4. Please add a conclusion section (with statements on challenges and perspectives).
5. Some of the Figure's quality can be enhanced, such as font size, resolution, etc.
Round 2
Reviewer 2 Report
The manuscript quality has been significantly improved after revision. Accept as is.